# Effect of Non-Isothermal Aging on the Mechanical Properties and Corrosion Resistance of 2A12 Aluminum Alloy

**DOI:** 10.3390/ma16113921

**Published:** 2023-05-24

**Authors:** Jie Yang, Hongfeng Liu, Tao Zeng, Shuai Li, Zhongying Liu, Tingting Wu, Dongdong Gu

**Affiliations:** 1School of Mechanical Engineering, North China University of Water Resources and Electric Power, Zhengzhou 450045, China; yangjie@ncwu.edu.cn (J.Y.); 17513362976@163.com (H.L.); liuzhongying87@126.com (Z.L.); ttabbmmjj@163.com (T.W.); wintergdd@126.com (D.G.); 2School of Materials Science and Engineering, Dalian University of Technology, Dalian 116024, China; zengtao1652@163.com

**Keywords:** 2A12 aluminum alloy, non-isothermal aging, mechanical properties, corrosion properties

## Abstract

The effect of linear non-isothermal aging and composite non-isothermal aging on the mechanical properties and corrosion resistance of 2A12 aluminum alloy was investigated. Optical microscopy (OM) and scanning electron microscopy (SEM) equipped with energy-dispersive spectroscopy (EDS) were used to study the microstructure and intergranular corrosion morphology, and the precipitates were analyzed using X-ray diffraction (XRD) and transmission electron microscopy (TEM). The results showed that the mechanical properties of 2A12 aluminum alloy were improved by both non-isothermal aging techniques due to the formation of an S′ phase and a point S″ phase in the alloy matrix. Linear non-isothermal aging resulted in better mechanical properties than composite non-isothermal aging. However, the corrosion resistance of the 2A12 aluminum alloy was reduced after non-isothermal aging due to the transformation of matrix precipitates and grain boundary precipitates. The corrosion resistance of the samples followed the order: annealed state > linear non-isothermal aging > composite non-isothermal aging.

## 1. Introduction

Due to their high specific strength, easy processing, and recycling, 2000-series aluminum alloys have historically played important roles in the defense, transportation, and aerospace industries [1,2]. The industry has higher requirements for materials, and 2000-series aluminum alloys are subjected to increasingly stringent requirements and must have high strength, corrosion resistance, and fracture toughness. To meet these requirements, reduce the influence of harmful elements, and improve the overall performance of materials, researchers have designed new alloy compositions, material preparation methods, and heat treatment methods [3,4]. Generally, the heat treatment process of 2000-series aluminum alloys mainly include homogenization, solution treatment, and aging [5]. Many investigations have been carried out to improve the comprehensive properties of 2000-series aluminum alloys [6,7,8].

The solid solution temperature and time are the most important parameters during solution treatment. Li et al. [9] studied the effect of the solid solution temperature on the microstructure and properties of 2A97 aluminum alloy and found that the plasticity decreased as the temperature increased. Lu et al. [10] investigated the effect of solution treatment on the mechanical properties and microstructure of an Al-Cu-Mg alloy containing Ag. The yield strength and tensile strength of the alloy first increased and then slowly decreased as the temperature increased.

The aging and strengthening processes of aluminum alloys include natural aging, single aging, double aging, three-step aging, and so on. The main aging parameters are aging temperature and aging time. Blankenship et al. [11] investigated the influence of aging temperature on an Al-Cu-Li alloy and found that aging promoted the precipitation of δ′ phase when the aging temperature was 125 °C. Zhang et al. [12] showed that δ′ (Al3Li) and θ′ (Al2Cu) phases were the main precipitates when the aging temperature was lower than 160 °C. However, the main precipitate was the T1(Al2CuLi) phase when the aging temperature was higher than 160 °C. Chen et al. [13] investigated the precipitate evolution of 2195 aluminum alloy with different aging temperatures and times and showed that the precipitate type, size, and rate all changed under different aging conditions. The above investigations mainly focused on one-step aging. To further improve the comprehensive performance of 2000-series aluminum alloys, researchers have developed multi-stage aging processes. Li et al. [14] found that the strength of an Al-Cu-Li alloy after double aging (high temperature, then low temperature) and triple aging (low temperature, high temperature, then low temperature) was higher than that of the T6 state. Wang et al. [15] studied the influence of intermittent aging on the corrosion resistance of a new Al-Cu-Mg-Ag alloy and found that intermittent aging increased the intergranular corrosion resistance and exfoliation corrosion resistance. Zhang et al. [16] investigated the effects of single aging and multi-stage intermittent aging on the microstructure and properties of Al-Cu-Mg-Ag-Zr alloys and found that multi-stage intermittent aging treatment improved the fracture toughness of the alloys while maintaining the strength, hardness, and conductivity compared with the T6 condition.

The above discussions only investigated the evolution of microstructure and properties in the isothermal state. In 2007, Staley [17] first proposed a non-isothermal aging process. Most research on non-isothermal aging has mainly focused on 7000-series aluminum alloys, whose thermodynamic parameters in non-isothermal environments are different from those in isothermal environments. Jiang et al. [18] found that the strength and corrosion resistance of Al-Zn-Mg-Cu were improved by non-isothermal heat treatment. Chen et al. [19] found that the non-isothermal aging of an Al-Zn-Mg-Cu alloy obtained better mechanical properties and corrosion resistance than that subjected to retrogression and re-aging. Li et al. [20] found that the main strengthening precipitates were GP zones and η′ phases during non-isothermal aging. An ultimate yield strength of 582 MPa was obtained when an Al-5.87Zn-2.07Mg-2.42Cu alloy was cooled to 140 °C. Guo et al. [21] showed that the corrosion resistance of 7N01 aluminum alloy improved after non-isothermal aging, and the maximum aging temperature (*T*max) significantly affected the mechanical properties. Li et al. [22] found that the corrosion resistance of Al-Zn-Mg alloy was improved when subjected to 5 min of air cooling, followed by water quenching after non-isothermal aging. Peng et al. [23] investigated the non-isothermal aging of 7050 alloy and found that the hardness and tensile strength improved to 182 HV and 578 MPa as the aging progressed. This was related to the growth and coarsening of intergranular precipitates during heating and the formation of fine intragranular precipitates during cooling. 

Most investigations of non-isothermal aging have mainly focused on 7000-series aluminum alloys, but similar studies on 2000-series aluminum alloys are rare. Both 7000-series and 2000-series aluminum alloys are categorized as aging-strengthening aluminum alloys. Thus, it is beneficial to investigate the impact of non-isothermal aging on the properties of the latter. The objective of this paper is to investigate the effect of non-isothermal aging on the corrosion resistance and mechanical properties of 2A12 aluminum alloy based on immersion corrosion tests and microstructure characterization. This will provide theoretical guidance for the development of new aging processes for 2000-series aluminum alloys.

## 2. Materials and Methods 

The chemical composition of 2A12 aluminum alloy is listed in Table 1, which was determined by X-ray fluorescence spectrometry. The 2A12 aluminum alloy was treated in a resistance heating furnace at a solution temperature of 498 °C for 30 min. After rapid water cooling, non-isothermal aging was carried out. The quenching transfer time was limited to 5 s as precisely as possible. Two different non-isothermal heat treatments, linear non-isothermal aging, and composite non-isothermal aging, were carried out, as shown in Figure 1a and Figure 1b, respectively.

The metallographic microstructure and intergranular corrosion features of 2A12 aluminum alloy obtained after different non-isothermal aging processes were obtained on a LeciaMEF-4 metallographic microscope. After grinding and polishing, the specimens were treated with Keller solution for 6–8 s. The solution volume ratio of Keller solution was HF:HCl:HNO_3_:H_2_O = 1:1.5:2.5:95. The fracture morphology of samples subjected to different thermal cycles was observed by scanning electron microscopy (SEM, Oberkochen, Germany). 

To analyze the precipitate evolution of 2A12 aluminum alloy after different non-isothermal heat treatments, X-ray diffraction (XRD), electron backscatter diffraction (EBSD), and transmission electron microscopy (TEM) were carried out. An EMPUREAN instrument was used for X-ray diffraction analysis to confirm the identity of the secondary phase. The instrument parameters were a working voltage of 35 kV, a detection angle range of 0–100°, and a scanning rate of 2°/min. EBSD specimens subjected to different thermal cycles were first ground and mechanically polished, then put into an electrolytic solution (30% HNO_3_ + 70% H_2_O) to polish at −30 °C for 8 s at a voltage of 15 V to avoid generating residual stresses on the surface. TEM specimens were carefully polished to a thickness of 40–70 µm and then disk-shaped samples with a diameter of 3 mm were obtained by a punching machine. Then, the samples were subjected to a double spray treatment on a TENUPOL-5 instrument. The corrosion solution volume ratio was HNO_3_:CH_3_OH = 1:9. Liquid nitrogen was added to cool the samples to −26 °C. The voltage was 15 V, and the current was 40 mA. The TEM analysis of the samples was performed on a TECNAG200S-TWIN.

### 2.1. Thermal Property Analysis

Differential scanning calorimetry (DSC) was carried out on a TA DSC Q100 to analyze the thermodynamic performance of 2A12 aluminum alloys after different thermal cycles. The test temperature was increased from 30 °C to 460 °C at a rate of 10 °C/min.

### 2.2. Tensile Test and Microhardness Analysis

Tensile tests were performed on an NDS 100 tensile tester at a tensile rate of 5 mm/min. The specific dimensions of the specimen are shown in Figure 2. To ensure the accuracy of the data, each group of specimens was tested at least three times. Microhardness tests for specimens subjected to different thermal cycles were performed on an HVS-1000Z machine. The test load was 0.5 kg with a duration of 15 s.

### 2.3. Corrosion Test

The intergranular corrosion (IGC) of 2A12 aluminum alloy was analyzed according to GB/T7998-2005 [24], and the dimensions were 20 mm × 20 mm × 1.4 mm. The samples subjected to different thermal cycles were first degreased with 10 wt.% NaOH, pickled in 30 vol.% HNO_3_, washed with acetone and deionized water, and dried with a hair dryer. The samples were immersed in a corrosive solution (57 g NaCl + 10 mL H_2_O_2_ (30%) diluted to 1000 mL) at 35 ± 2 °C for 6 h in a thermostatic water bath. At least three samples were analyzed for each corrosion test. 

The exfoliation corrosion test of 2A12 aluminum alloy subjected to different thermal cycles was carried out based on the GB/T22639-2022 standard [25]. Samples were prepared with dimensions of 20 mm × 20 mm × 1.4 mm. The exfoliation corrosion solution preparation method was as follows: 4 M NaCl, 0.5 M KNO_3_, and 0.1 M HNO_3_ were mixed and diluted to 1000 mL with deionized water. Then, the specimens were immersed in the corrosive solution at 25 ± 2 °C for 96 h in a thermostatically controlled water bath. The macroscopic corrosion morphology of the specimens was recorded using photos at 12 h, 24 h, 48 h, 72 h, and 96 h.

Electrochemical experiments were performed using a CS350 electrochemical workstation, and a three-electrode system was used to study the electrochemical corrosion of samples under different heating conditions. The sample was used as the working electrode, and an Ag/AgCl electrode and a Pt electrode were used as the reference electrodes. The electrochemical samples were ground to 2000# by SIC paper, then washed with alcohol and deionized water. After 1 h in the solution, the open-circuit potential stabilized, and then the polarization curves were obtained. The test range was ±200 mV relative to the open-circuit potential at a scanning speed of 1 mV/s. All tests were performed at room temperature in a 1.0 M NaCl solution, and the exposed area was 1 cm^2^.

## 3. Results and Discussion

### 3.1. Microstructure

Figure 3 shows the metallographic morphology of 2A12 aluminum alloy after non-isothermal aging. The microstructure of the annealed and non-isothermal specimens was different. The microstructure of the non-isothermal samples in Figure 4 consisted of equiaxed grains, and the bulk or blocky microstructure with different sizes and an uneven distribution was the AlCuFeMnSi phase, which is often observed in aluminum alloys [26,27]. Many investigations have shown that the existence of the AlCuFeMnSi phase is detrimental to the mechanical properties and corrosion resistance [26,27]. Since the melting point of the AlCuFeMnSi phase is higher than the annealing temperature used in this investigation, the microstructure and morphology of these particles did not evolve after different heat treatments. No obvious difference in micromorphology was observed between linear non-isothermal aging and composite non-isothermal aging.

The XRD patterns of 2A12 aluminum alloy after different heating conditions are shown in Figure 5. The second phase contained in the 2A12 aluminum alloy after non-isothermal aging and after annealing was identical. The four diffraction peaks with the highest diffraction intensity all corresponded to aluminum. When the test angle was small, only the Al_2_CuMg phase was detected in the alloys under different heat treatments. The XRD peak intensity increased after annealing, indicating that more Al_2_CuMg phases were present in the 2A12 aluminum alloy. The Al_7_Cu_2_Fe phase was detected in all three samples, which exerts no strengthening effect. Compared with the annealed 2A12 aluminum alloy, the diffraction peaks after both non-isothermal aging treatments shifted to larger angles, indicating the occurrence of lattice distortion related to the formation of a transition phase after non-isothermal aging. Additionally, the dispersed second-phase particles Al_20_Cu_2_Mn_3_ (T phase) [28,29] were well-distributed throughout the substrate.

Figure 6 shows the DSC curve of the samples after different annealing treatments, in which two strong endothermic peaks (A, C) and a small endothermic peak were found, along with three strong exothermic peaks (B, D, and F). Peak A corresponds to the dissolution of the GP zones from 50 °C to 120 °C [30]. Similarly, peak B, with a wide temperature range (120–170 °C), represents the precipitation of the S″ phase. Peak C (170–200 °C) was related to the re-dissolution of the S″ phase. The exothermic peak D (215 °C) was associated with the formation of the S′ phase, while peak E (225 °C) indicated the partial re-dissolution of the S′ phase. The precipitation of the S″ phase increased during non-isothermal aging, as represented by the broadening of peak D. The weaker peak D under both non-isothermal aging conditions indicated the formation of a less active S′ phase. The fluctuations in peak E were less obvious, indicating that less of the S′ phase redissolved and most remained in the matrix of 2A12 aluminum alloy. Finally, peak F (225–250 °C) with a high intensity represented the precipitation of the S phase, which is stable at room temperature. Therefore, more heat was released when the S′ phase was converted into the S phase, resulting in high intensity for peak F. The DSC curves of linear non-isothermal aging and composite non-isothermal aging samples show that the F curve of the latter curve was wider. This indicates that the precipitation temperature range of the S phase was wider and the size of the S phase precipitates was larger. The precipitation and dissolution temperature ranges of the GP zones, S″ phase, S′ phase, and S phase overlapped. When the former dissolved, the latter may have precipitated, which generated overlapping endothermic and exothermic peaks, ultimately forming an undulating DSC curve.

Figure 7 shows the crystal orientation map of 2A12 aluminum alloy with different thermal cycles by EBSD analysis. Figure 7a shows the orientation of the annealed grains. The fine annealed grains were arranged into fibers with a size range of 5–80 μm. Figure 7b shows the grain orientation after linear non-isothermal aging. Compared with the annealed alloys, the grain size was larger, and the grains were oriented perpendicular to the rolling direction. Many finely recrystallized grains were distributed at grain boundaries, and the grain size was 5–100 μm. Figure 7c shows the grain orientation after composite non-isothermal heat treatment. Compared with the annealed alloy, the grain size was larger, and the grains were oriented along the rolling direction.

Figure 8 shows the matrix precipitate morphology of the 2A12 aluminum alloy after annealing and non-isothermal heat treatment. The matrix precipitates in the annealed samples were mainly composed of rod-shaped dispersed T phases (Al_20_Cu_2_Mn_3_) and needle-shaped precipitated S phases (Al_2_CuMg), as displayed in Figure 8a. Figure 8b shows the micromorphology of the 2A12 aluminum alloy after linear non-isothermal aging. The acicular S′ phase and punctate S″ phase were mixed and distributed throughout the aluminum alloy matrix. In the non-isothermal heat treatment specimen, many dislocations appeared, which improved the mechanical properties of the alloy. Composite non-isothermal heat treatment produced a larger S′ phase and a more punctate S″ phase due to the longer aging time and lower aging temperature. This is consistent with the conclusion obtained from the DSC analysis shown in Figure 8c.

Figure 9 shows the morphology of grain boundary precipitates in the 2A12 aluminum alloy after different annealing conditions. A small amount of T phase formed at grain boundaries in the annealed 2A12 aluminum alloy. For the linear non-isothermal aged samples, only a small amount of T phase (Al_20_Cu_2_Mn_3_) and punctate precipitates were distributed at grain boundaries. There were more punctate precipitated phases near the grain boundary of the 2A12 aluminum alloy after composite non-isothermal aging. No precipitate-free zone was observed in the annealed sample or in the non-isothermal heat-treated specimens.

### 3.2. Mechanical Properties

#### 3.2.1. Tensile Properties

Figure 10 shows the tensile properties of the 2A12 aluminum alloy after annealing and different non-isothermal aging treatments. The tensile strength and elongation of the annealed alloy were 176 MPa and 25.6%, respectively. Compared with the annealed alloy, non-isothermal aging significantly improved the tensile strength of the 2A12 aluminum alloy. However, the elongation of the heat-treated specimens was lower. The tensile strength of the 2A12 aluminum alloy after linear non-isothermal heat treatment and composite non-isothermal heat treatment was 447 MPa and 445 MPa, respectively. Non-isothermal heat treatment improved the tensile strength of 2A12 aluminum alloy and decreased its elongation. The 2A12 aluminum alloy, after linear non-isothermal heat treatment, showed good comprehensive mechanical properties.

Figure 11 shows the fracture morphology of the 2A12 aluminum alloy under different heat treatment conditions. The fracture morphology indicated typical plastic fracture characteristics with many dimples [31]. Many inclusions were observed in the non-isothermal heat-treated specimens, which may have decreased the elongation of the 2A12 aluminum alloy. The fracture characteristics were consistent with the evolution of the mechanical properties of the 2A12 aluminum alloys at different heating conditions.

#### 3.2.2. Microhardness

Figure 12 shows that the Vickers microhardness of the annealed alloy, linear non-isothermal aging, and composite non-isothermal aging were 52 HV, 131 HV, and 122 HV, respectively. Non-isothermal aging improved the microhardness of the 2A12 aluminum alloy. The evolution of mechanical properties in the aging-strengthened aluminum alloy was mainly related to the size and distribution of precipitates. During non-isothermal aging, precipitates form upon changing the annealing temperature. The types and contents of the strengthening phases were different depending on the thermal treatment, and the strengthening effect was also different. Additionally, the strength and hardness of the alloy were significantly increased due to the entanglement of dislocations and precipitated phases [32].

### 3.3. Corrosion Performance

#### 3.3.1. Intergranular Corrosion

Figure 13 shows the intergranular corrosion morphology of 2A12 aluminum alloy after different heat treatments. Figure 13a shows the intergranular corrosion morphology of the annealed alloy. Grain boundary corrosion was not observed, and the maximum corrosion depth was 103.6 μm. The intergranular corrosion depths of linear non-isothermal aging and composite non-isothermal aging were 109.0 μm and 123.8 μm, respectively. The intergranular corrosion morphologies of 2A12 aluminum alloy after non-isothermal aging were similar, and both showed typical network-like intergranular corrosion characteristics. For the heat-treated aluminum alloys, local corrosion was primarily related to the potential difference between the matrix and grain boundaries. The grain boundaries always act as the anode and are therefore attacked first during initial immersion [26]. Overall, the intergranular corrosion resistance of the non-isothermal aging specimens was lower than that of the annealed alloy, and the intergranular corrosion resistance after linear non-isothermal aging was better than after composite non-isothermal aging. According to the maximum intergranular corrosion depth of the three samples under different heat treatment conditions, the intergranular corrosion grades of the alloys after annealing, linear non-isothermal aging, and composite non-isothermal aging were all classified as grade 4 according to the GB/T7998-2005 standard [24].

For the linear non-isothermal aged sample, the acicular S′ phase and the punctate S″ phase were mainly formed in the matrix, with few grain boundary precipitates. No precipitate-free zone was observed. For the composite non-isothermal-aged specimen, the size of the S′ phase and the number of S″ phases increased due to the longer duration of the thermal cycle. This increased the potential difference between the matrix and grain boundaries. Consequently, the composite, non-isothermal aged specimen had the lowest corrosion resistance. The annealed 2A12 aluminum alloy contained few grain boundary precipitates, and no continuous corrosion channels were observed. Therefore, it displayed the best intergranular corrosion resistance among the three different heating conditions [33]. The appearance of pitting (a black area) was associated with the formation of the AlCuFeMnSi phase [34]. Specifically, a corrosive galvanic cell was formed between the aluminum matrix and AlCuFeMnSi, and anodic dissolution occurred in the aluminum matrix around the AlCuFeMnSi phase.

#### 3.3.2. Exfoliation Corrosion

Figure 14 shows the exfoliation corrosion morphology of 2A12 aluminum alloy after different heat treatment conditions. Pitting corrosion occurred first, and extensive corrosion finally developed into exfoliation corrosion.

At the beginning of corrosion, three samples showed pitting corrosion when the immersion time was less than 12 h. The annealed samples contained fewer and smaller pits. There were more pits in the sample subjected to linear non-isothermal aging, but the pits were shallow. Deeper and denser pits were observed in the samples after composite non-isothermal aging. It can be tentatively speculated that non-isothermal aging reduced the exfoliation corrosion resistance of 2A12 aluminum alloy, and the exfoliation corrosion resistance after linear non-isothermal aging was better than that after composite non-isothermal aging. As the immersion time progressed to 24 h, the pitting in the annealed specimens became deeper, and local pitting became more severe. For the linear non-isothermal aged specimen, the pitting density increased with the immersion time, and slight exfoliation corrosion appeared for the composite non-isothermal aged sample. Upon increasing the immersion time, exfoliation corrosion occurred in both the annealed and non-isothermal specimens. The volume of corrosion products at grain boundaries exceeded that of the corroded metal, resulting in wedging stress and the lifting of surface grains [26].

In general, the exfoliation corrosion resistance of the samples after non-isothermal aging decreased, and the specimen subjected to composite non-isothermal aging displayed the lowest exfoliation corrosion among the three different samples. According to the GB/T 22639-2022 standard [25], the exfoliation corrosion levels of the specimens after annealing, linear non-isothermal aging, and composite non-isothermal aging were evaluated as EA, EB, and EC grades, respectively. The variation of the exfoliation corrosion grade of the alloy with immersion time is shown in Table 2.

#### 3.3.3. Electrochemical Analysis

Figure 15 shows changes in the open-circuit potential (OCP) with time for 2A12 aluminum alloy after different heat treatment conditions. When the test time reached 3600 s, the OCP stabilized. The OCP of annealed 2A12 aluminum alloy was the highest, followed by the linear non-isothermal aged specimen. The OCP of the composite non-isothermal sample was the lowest.

After the OCP stabilized, the potentiodynamic polarization curves of the samples were recorded, as shown in Figure 16 and Table 3. Generally, the higher the corrosion current density, the lower the corrosion resistance. Thus, it can be concluded that the corrosion resistance followed the order: annealed state > linear non-isothermal aging > composite non-isothermal aging. The results were consistent with the results of the intergranular corrosion and exfoliation corrosion experiments, indicating that the curve fitting from the polarization curves could be used to calculate the corrosion current density (*i*_corr_) of the heated-treated 2A12 aluminum alloy, although the fitting process may be subjective.

The *E*_corr_ values of the non-isothermal-aged specimens were more positive than those of annealed-state alloys. The cathodic current densities of the specimens after different heat treatment conditions increased slightly below *E*_corr_. However, the anodic current densities of the alloys increased rapidly at a potential just above *E*_corr_, as displayed in Figure 16. This was related to pitting corrosion. The rate of increase in the anodic current densities of the alloys gradually slowed upon increasing the potential and then gradually stabilized. Changes in the anodic current densities may have been related to the formation of a salt film during pitting corrosion, which inhibited mass transport processes [26].

## 4. Conclusions

The effect of non-isothermal aging on the evolution of the microstructure and properties of 2A12 aluminum alloy was investigated. The mechanical properties of 2A12 aluminum alloy were improved after non-isothermal aging, which was associated with the formation of the S′ phase and the point S′′ phase in the matrix. Specifically, linear non-isothermal aging provided better mechanical properties than composite non-isothermal aging. Non-isothermal aging deteriorated the corrosion resistance of 2A12 aluminum alloy, which followed the order: annealed state > linear non-isothermal aging > composite non-isothermal aging. The evolution of the corrosion susceptibility of the heat-treated specimens was caused by changes in the potential difference between the matrix and grain boundaries. The negative effect of non-isothermal aging on the corrosion susceptibility of 2A12 aluminum alloy must be considered.

## Figures and Tables

**Figure 1 materials-16-03921-f001:**
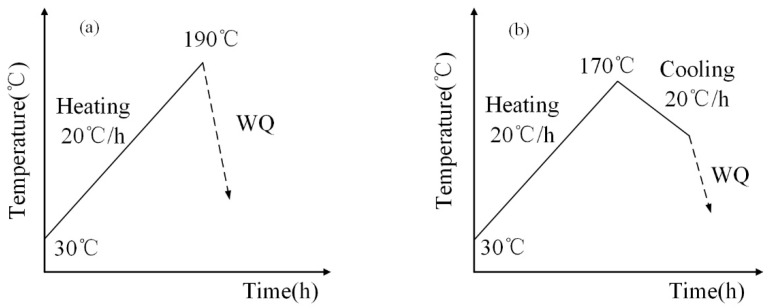
Schematic diagram of the non-isothermal aging process (**a**) 20 °C/h ((30, 190) × 20) (**b**) 20 °C/h ((30, 190) × 20 + (170, 100)/20).

**Figure 2 materials-16-03921-f002:**
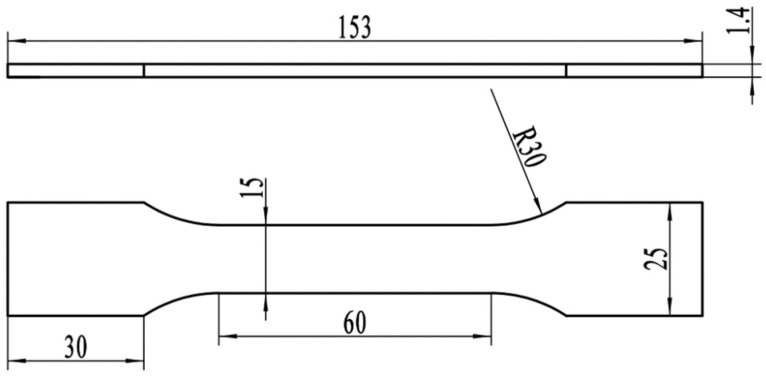
Dimensions of the sample for the tensile test (mm).

**Figure 3 materials-16-03921-f003:**
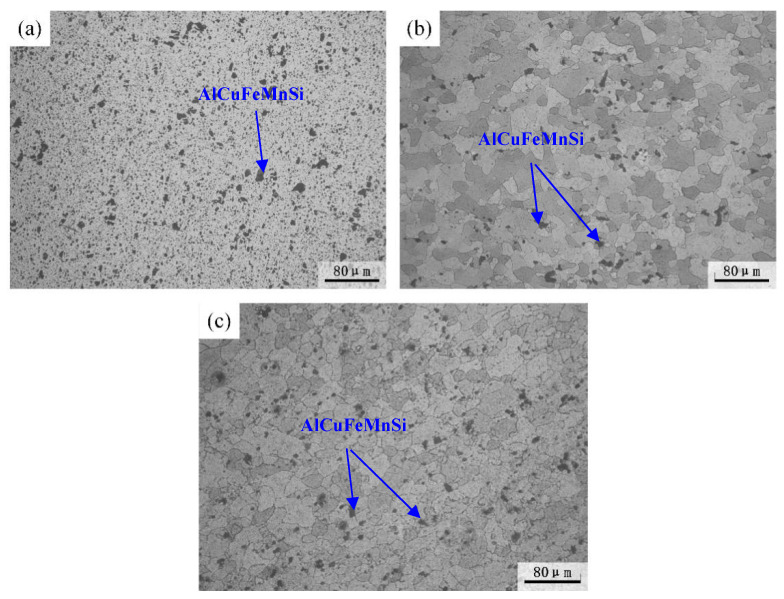
Metallographic structure of 2A12 aluminum alloy under different conditions: (**a**) Annealed state; (**b**) (30,190) × 20; (**c**) (30,170) × 20 + (170,100)/20.

**Figure 4 materials-16-03921-f004:**
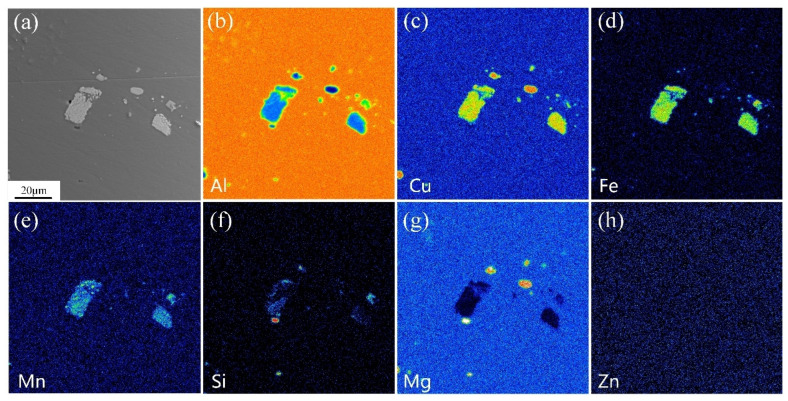
Element distribution maps of the inclusion phase of the non-isothermally aged 2A12 aluminum alloy. (**a**) SEM picture; (**b**) Al; (**c**) Cu; (**d**) Fe; (**e**) Mn; (**f**) Si; (**g**) Mg; (**h**) Zn.

**Figure 5 materials-16-03921-f005:**
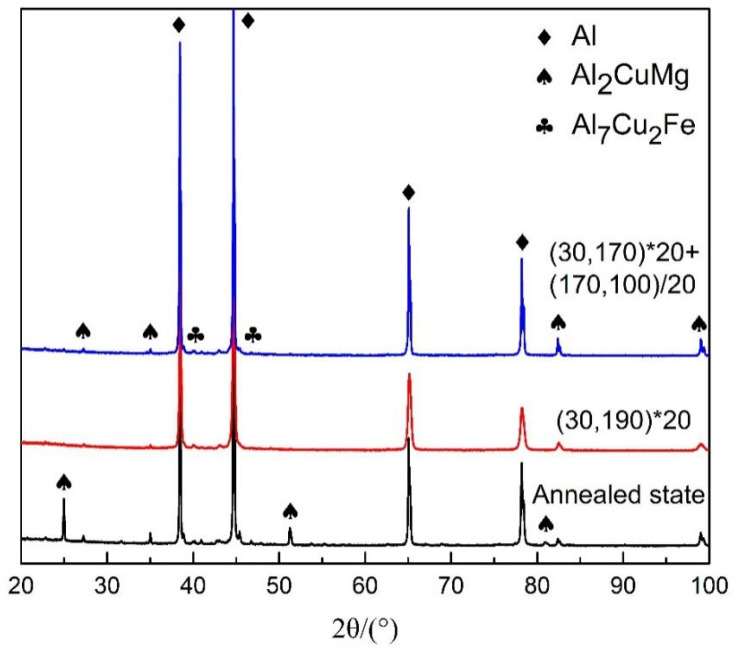
XRD pattern of 2A12 aluminum alloy after different aging conditions.

**Figure 6 materials-16-03921-f006:**
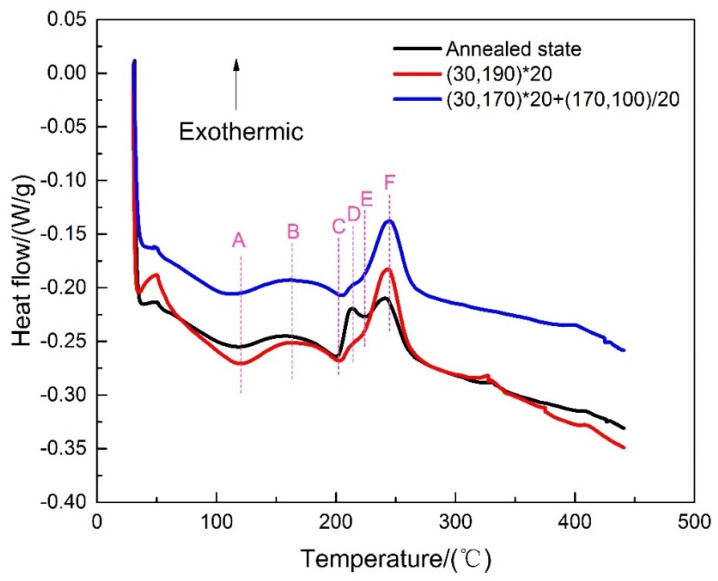
DSC curve of 2A12 aluminum alloy after different annealing conditions.

**Figure 7 materials-16-03921-f007:**
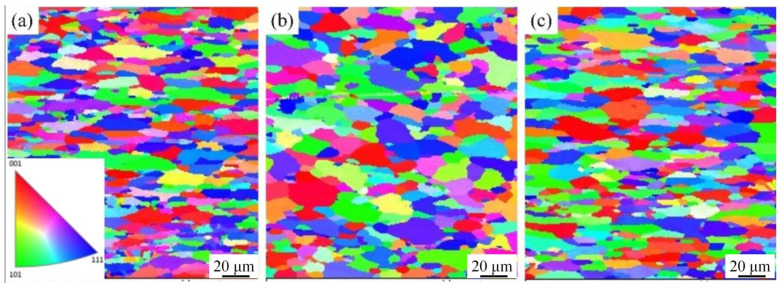
Grain orientation of 2A12 aluminum alloy in the (**a**) annealed state; (**b**) (30,190) × 20; (**c**) (30,170) × 20 + (170,100)/20.

**Figure 8 materials-16-03921-f008:**
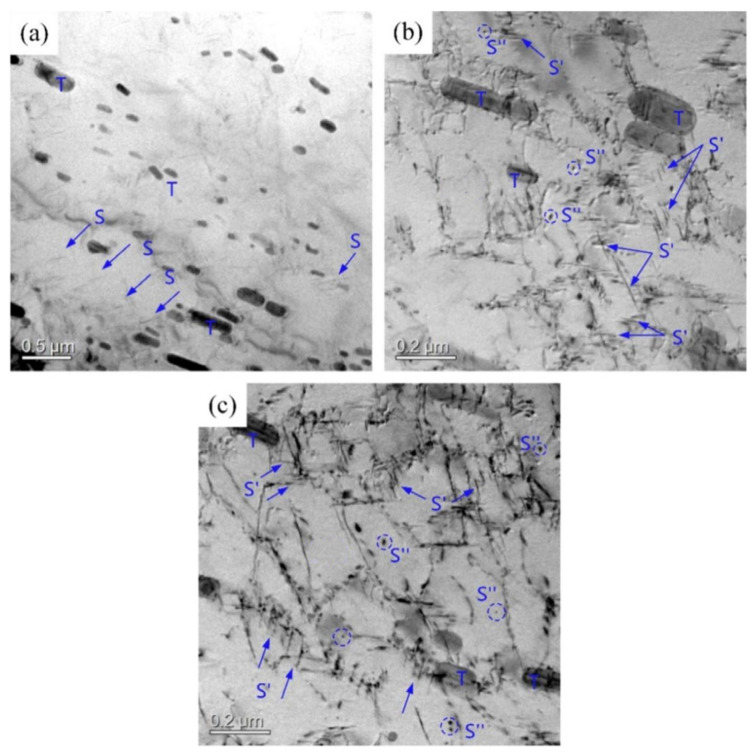
TEM morphology of 2A12 aluminum alloy under different conditions: (**a**) Annealed state; (**b**) (30,190) × 20; (**c**) (30,170) × 20 + (170,100)/20.

**Figure 9 materials-16-03921-f009:**
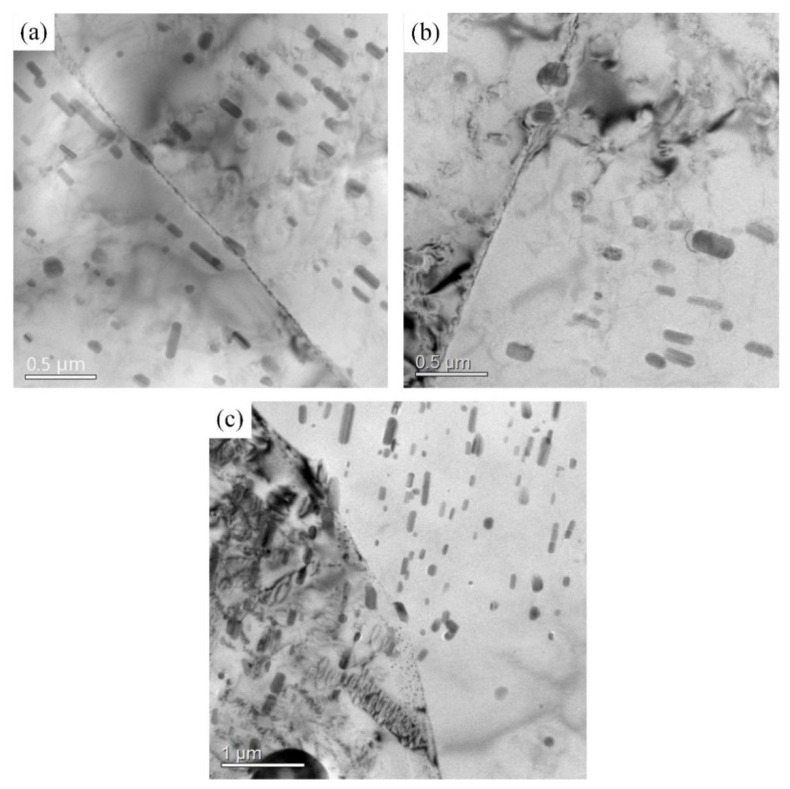
Grain boundary TEM morphology of 2A12 aluminum alloy under different conditions: (**a**) annealed state; (**b**) (30,190) × 20; (**c**) (30,170) × 20 + (170,100)/20.

**Figure 10 materials-16-03921-f010:**
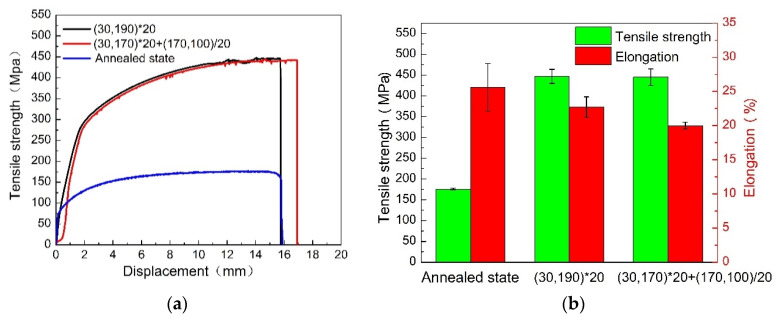
Evolution of mechanical properties of 2A12 aluminum alloy under different conditions: (**a**) typical tensile stress-displacement curves, (**b**) tensile strength and elongation of specimens.

**Figure 11 materials-16-03921-f011:**
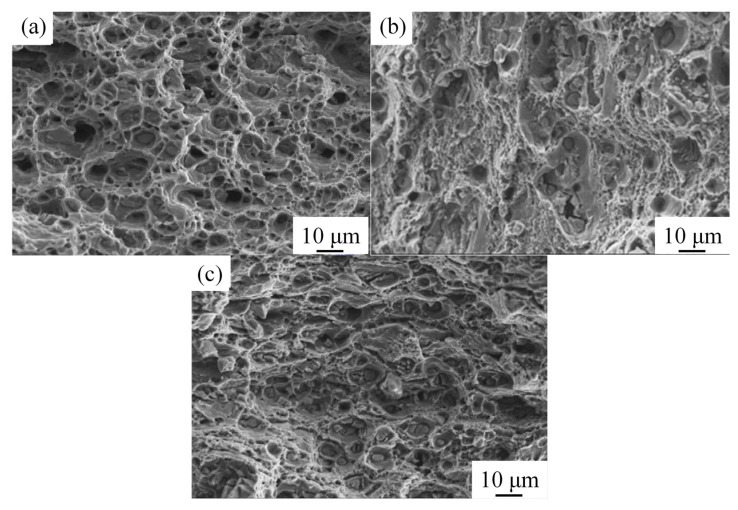
Fracture morphology of 2A12 aluminum alloy under different conditions for the (**a**) annealed state; (**b**) (30,190) × 20; (**c**) (30,170) × 20 + (170,100)/20.

**Figure 12 materials-16-03921-f012:**
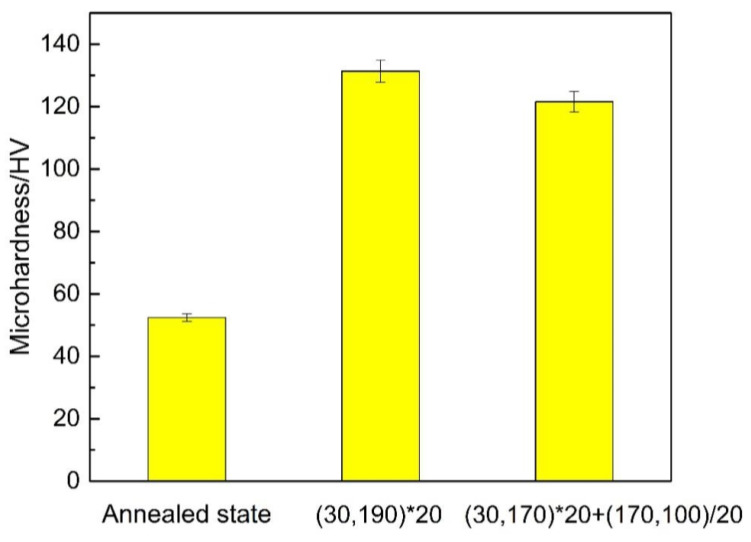
Vickers hardness of 2A12 aluminum alloy under different conditions.

**Figure 13 materials-16-03921-f013:**
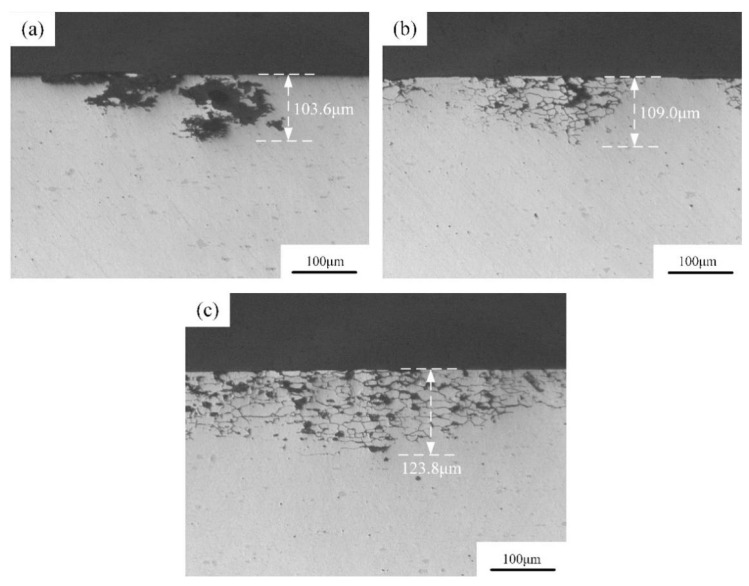
Intergranular corrosion morphology of 2A12 aluminum alloy after different heat treatments: (**a**) Annealed state; (**b**) (30,190) × 20; (**c**) (30,170) × 20 + (170,100)/20.

**Figure 14 materials-16-03921-f014:**
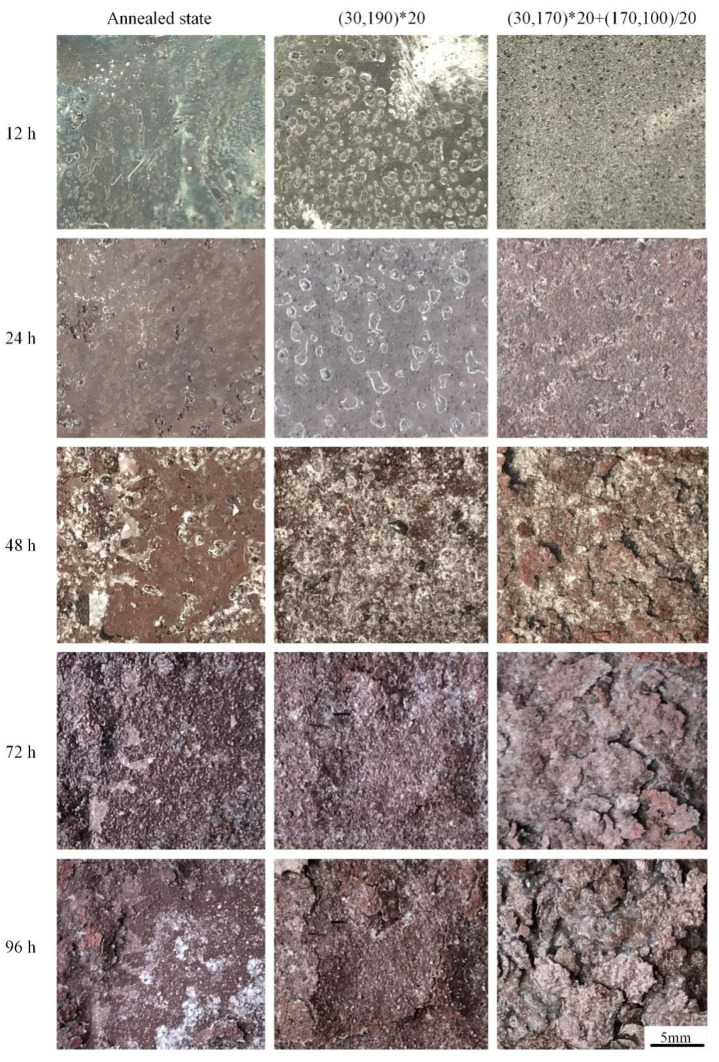
Exfoliation corrosion morphology of 2A12 aluminum alloy after different heat treatment conditions.

**Figure 15 materials-16-03921-f015:**
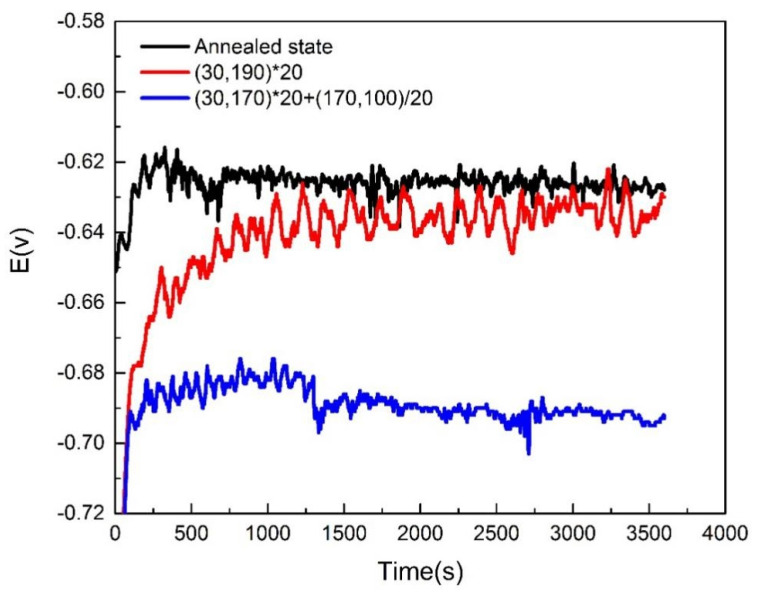
Open-circuit potential of 2A12 aluminum alloy after different heat treatments.

**Figure 16 materials-16-03921-f016:**
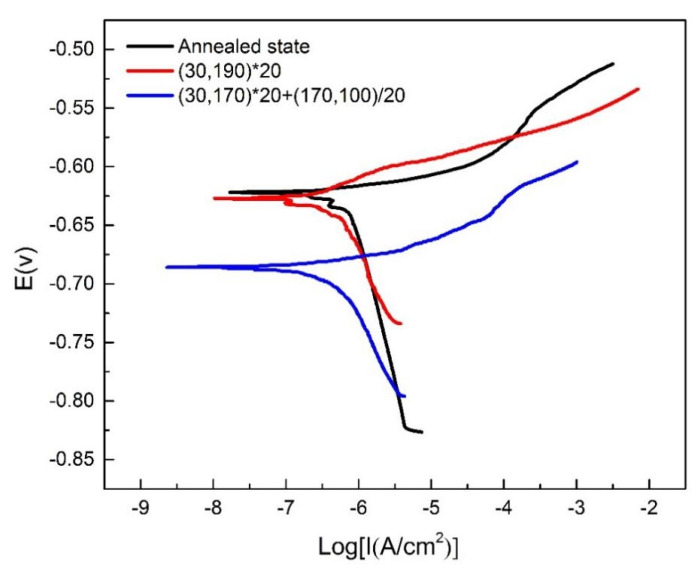
Potential dynamic polarization curves of 2A12 aluminum alloy after different heat treatment conditions.

**Table 1 materials-16-03921-t001:** Chemical composition of 2A12 aluminum alloy (wt.%).

Element	Cu	Mg	Mn	Fe	Si	Zn	Ti	Ni	Other	Al
Contents	4.64	1.59	0.59	0.42	0.32	0.18	0.1	0.05	0.15	Bal.

**Table 2 materials-16-03921-t002:** Exfoliation corrosion level changes with corrosion time for 2A12 aluminum alloy under different conditions.

Conditions	Immersion Time				
	12 h	24 h	48 h	72 h	96 h
Annealed state	PA	PB	PC	EA	EA
(30,190) × 20	PA	PB	PC	EA	EB
(30,170) × 20 + (170,100)/20	PB	PC	EB	EC	EC

**Table 3 materials-16-03921-t003:** Electrochemical corrosion parameters of 2A12 aluminum alloy under different conditions.

Sample	*E*_corr_ (mV (Ag/AgCl))	*I*_corr_ (mA/cm^2^)
Annealed state	−621.95	2.2386 × 10^−7^
(30,190) × 20	−627.37	2.3613 × 10^−7^
(30,170) × 20 + (170,100)/20	−685.85	4.3455 × 10^−7^

## Data Availability

The data that support the findings of this study are available from the corresponding author upon reasonable request.

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
