# Peer review of "Effect of Non-Isothermal Aging on the Mechanical Properties and Corrosion Resistance of 2A12 Aluminum Alloy"

_materials, 2023, doi:10.3390/ma16113921_

Round 1

Reviewer 1 Report

This manuscript investigate the effect of non-isothermal aging on the physical properties of 2A12 alloys. The authors consider two aging methods (linear non-isothermal aging and composite non-isothermal aging). They found that the linear non-isothermal process provides better mechanical properties for the sample. However, these process deteriorates the corrosion resistance. This manuscript present enough results and physical discussion in order to be published and deserves publication. However, I do recommend a few points to be improved before publication.

1) The last paragraph of introduction section should give the reader a more detailed explanation of what will be carried out in the manuscript, pointing out why this research is new and what improvement in the field is being done. Therefore, I recommend the authors to clarify those points in the introduction section.

2) The mechanical properties of the samples are slightly different. Could this be atributed random deffects of the samples? Shouldn't the authors compare the mechanical properties of several samples in order to confirm the better mechanical properties of linear non-isothermal aging?

3) The authors rank the corrosion resistance of the alloys. However, a more detailed physical discussion on  why this happen should be given.

in conclusion, the manuscript presents neu physics to be published but requires some revision before publication on Materials.

In the last paragraph of introduction section the phrase: "Compared to 7000 series aluminum alloy, 2000 series aluminum alloy has not so much heat treatment system" should be improved.

I advise the authors to review the english throughout the manuscript for typos and grammar mistakes.

Author Response

See the attachment for details.

Reviewer 2 Report

The manuscript presents an interesting study about the effect of non-isothermal aging on the mechanical properties and corrosion resistance of 2A12 aluminum alloy. However, the paper needs major revisions before it is processed further, some comments follow:

 Abstract

The abstract must be improved. The abstract must highlight the novelty of this study, to present the methods used to characterize the material and to contain the main quantitative and qualitative conclusions.

Introduction

Are affirmation which has not background in the literature. Please add citation for the fallows affirmations/paragraph: lines 26-36.

In the last paragraph of the Introduction section must be highlighted the novelty of this study, also must be write a shortly paragraph regarding the tests do it in this study.

 Experimental

Please change the name into Materials and Methods.

The chemical composition of the alloy was given by the manufacturer or the authors determinate? Please specify this aspect.

Change the name of 2.1. subsection into Microstructural characterization.

Introduce the work electrode surface area exposed and also the scanning rate.

 Results and discussion

Figure 3. Please add figure labels in order to highlight the interest zones for the reader.

Figure 4. It is preferable to used different colors for each element.

Also, extend the discussion about figures 3 and 4.

The discussion regarding the XRD results are poor. The authors must present also the crystalline planes for the significant peaks.

It is not enough to present just the values of Ecorr and Icorr, the authors must calculate and discuss the values of polarization resistance and corrosion rate.

Figures 4, 5, 6, 7, 8, and 9 are not clear. Please replace them.

Regarding the electrochemical tests, it is not enough to discuss just current density and corrosion potential. The authors must calculate, introduce and discuss the polarization resistance and corrosion rate values. Also, this discussion must be compared with other studies.

Conclusions

The conclusion section must be improved. Add some suggestions and limitations. Also, add some quantitative results.

Author Response

See the attachment for details.

Reviewer 3 Report

The paper is studying the Effect of non-isothermal aging on the mechanical properties and corrosion resistance of a conventional 2A12 aluminum alloy. The topic is interesting however there is some questions in the manuscript need to be answered or corrected:

1) Fig. 4 which isothermal condition.

2) Fig. 8 the author mentioned substructures and deformation proportion. I don’t know why investigate the percentage if substructures without mentioning the preparation process of the alloy.

3) Fig. 9 and 10 how authors identified the second phase types. The authors need to mention the methods used and it should be at least two different methods to confirm such phases.

4)  Fig. 11 header (a) and (b) are missing and why the rate of stress strain of the red color condition is different from others. How many samples measured for tensile test for each condition?

5) the current manuscript discussion is very weak.

the current English Language is acceptable but can be improved. several sentences need to written in clear form. for instance: 'The above discussions belong to the evolution of microstructure and property in the 67 process of isothermal condition.' 

Author Response

See the attachment for details.

Round 2

Reviewer 3 Report

the authors did many corrections, Yet I only recommend to add more references related to the discussion part. after that, I agree to publish the current research work

the English is improved but I still recommend authors to revise it again as some sentences are hard to understand

Author Response

Ms. Ref. No.:  materials-2373989

Title: Effect of non-isothermal aging on the mechanical properties and corrosion resistance of 2A12 aluminum alloy

Materials

Dear Editor,

Thank you very much for your useful comments and suggestions on our manuscript. We have revised the manuscript accordingly, and the detailed revisions are listed below point by point.

The authors did many corrections, Yet I only recommend to add more references related to the discussion part. after that, I agree to publish the current research work

  • We have added add more references related to the discussion part in the revised manuscript, the details are displayed in color text.

The English is improved but I still recommend authors to revise it again as some sentences are hard to understand.

  • The manuscript has been carefully edited by a native English-speaking editor of MogoEdit, and the grammar, spelling, and punctuation have been verified and corrected where needed. Based on this review, we believe that the language in this paper meets academic journal requirements.

The revised manuscript has been resubmitted to your journal. We look forward to your response.

Sincerely,
Dr. Shuai Li

School of Mechanical Engineering
North China University of Water Resources and Electric Power
36 Beihuan Road, Jin Shui Zone
Zhengzhou City 450045, P R China
Tel: 86 371 69127295
Fax: 86 371 69127295

Email: lyctlishuai@163.com
